# Minimally Invasive Non-Surgical Technique in the Treatment of Intrabony Defects—A Narrative Review

**DOI:** 10.3390/dj11010025

**Published:** 2023-01-11

**Authors:** Styliani Anoixiadou, Andreas Parashis, Ioannis Vouros

**Affiliations:** 1Department of Periodontology, School of Dental Medicine, Aristotle University, 54124 Thessaloniki, Greece; 2Private Practice, 11643 Athens, Greece

**Keywords:** minimally invasive non-surgical technique, intrabony defects, periodontal treatment

## Abstract

Intrabony defects occur frequently in periodontitis and represent sites that, if left untreated, are at increased risk for disease progression. Although resective or repair procedures have been used to treat intrabony defects, aiming at their elimination, the treatment of choice is surgical periodontal regeneration. The development of periodontal regeneration in the last 30 years has followed two distinctive, though totally different, paths. The interest of researchers has so far focused on regenerative materials and products on one side, and on novel surgical approaches on the other side. In the area of materials and products, three different regenerative concepts have been explored namely, barrier membranes, bone grafts, and wound healing modifiers/biologics, plus many combinations of the aforementioned. In the area of surgical approaches, clinical innovation in flap design and handling, as well as minimally invasive approaches, has radically changed regenerative surgery. Recently, a minimally invasive non-surgical technique (MINST) for the treatment of intrabony defects was proposed. Initial clinical trials indicated comparable results to the surgical minimally invasive techniques in both clinical and radiographic outcomes. These results support the efficacy of this treatment approach. The aim of this review is to present the evidence on the application of minimally invasive non-surgical techniques and their efficacy in the treatment of intrabony defects.

## 1. Introduction

The treatment of intrabony periodontal defects has always proven to be challenging during the treatment of periodontal patients. A review of treatment modalities will be presented, with a focus on newer more conservative non-surgical approaches that have been suggested in recent years.

Periodontal intrabony defects are osseous defects with a specific morphology. The bottom of these defects is located more apically than the alveolar crest and they are surrounded by bony walls on 1-, 2-, or 3- sides, with the tooth root forming an additional wall. The prevalence of these defects is significantly lower than supracrestal periodontal defects.

Nielsen et al. [1] published data from a population of 209 subjects 15–70 years old who presented for dental treatment. Patients were examined clinically and radiographically. The results showed that 23% of the participants presented periodontal lesions, while 18% had at least one intrabony defect. The frequency of intrabony defects occurrence was associated with age. A total of 37% of the subjects older than 60 years old presented one or more intrabony defects, while in ages between 30 and 44 years old, defect occurrence was 18%. Wouters et al. [2] examined with a ×5 magnification, radiographs of a randomly selected sample of 733 subjects, over 20 years old. At least one intrabony defect occurred in 32% of the population group, with this percentage significantly being related to the subjects’ age.

In 1988, Papapanou et al. [3] published data from radiographic examination of subjects presenting for dental treatment in a university dental clinic. The sample of the research was 531 participants between 25 and 75 years old and the results showed that 8% of the teeth exhibited an intrabony defect. The frequency of intrabony defects was positively related to the subjects’ age. More correlations regarding intrabony defect location were reported in the study of Vrotsos et al. [4]. The authors collected data from 286 patients with moderate or advanced adult periodontitis scheduled for periodontal surgery. A sample of 5476 teeth was examined during osseous surgery and classified according to the bony walls presented. Intrabony defects were detected in 17.9% of the teeth included (981 out of 5476 teeth). Regarding defect location, in the maxilla, 15.4% of the teeth presented an intrabony defect, while in the mandible the corresponding percentage was 22.4% of the teeth. The posterior segment of the mandible had the highest prevalence of intrabony defects (33.8%) followed by the posterior maxillary segment (19.9%). The prevalence of the defects in the anterior segment did not present any differences between the two arches.

In a more recent publication, Najim et al. [5] presented data from a radiographic examination of 329 subjects, aged between 40 and 70 years old. Intrabony defects were detected in 2.2% of the teeth and were positively related to the age and oral health of the subject, as this was evaluated from the plaque score.

All the aforementioned studies are characterized by a lack of homogeneity, which may be due to different inclusion criteria, age limits, and periodontal conditions of the included population. The prevalence of intrabony defects fluctuates between 18 and 32% at the patient level [1,2], and 2.2 and 17.9% at the tooth level [3,4,5]. Whether or not these defects increase the risk for disease progression is not completely clarified. It has been suggested that if intrabony defects are left untreated, the probability of long-term loss of the involved tooth is highly increased. Papapanou et al. [6], in a 10-year follow-up, studied the incidence of tooth loss in teeth with untreated periodontal defects. The authors examined 201 subjects at two different timepoints (baseline and 10 years later), who did not receive any periodontal therapy. The bone defects detected in the participants were categorized according to their intrabony depth (1:2 mm, 2:2.5–4 mm, 3: ≥4.5 mm). The results showed the presence of intrabony defects in 9% of the teeth, with 67% of them being shallow defects (category 1). The progression of periodontal disease, as documented by tooth loss, was more pronounced in teeth with deeper lesions than in those with shallower defects. In the 10-year follow-up, only 13% of the teeth with supracrestal defects were lost, while the percentage of lost teeth with intrabony defects was 22%, 46%, and 68% in teeth with shallow, moderate, and deep lesions, category 1, 2, and 3 defects, respectively. Thus, the authors concluded that deeper intrabony lesions had increased percentages of tooth loss if periodontal treatment was not administered.

However, the treatment of intrabony defects led to periodontal stability and only minor progression seemed to occur if patients complied with supportive periodontal therapy. According to Pontoriero et al. [7], the progression of periodontal disease was similar in subjects with supracrestal and intrabony lesions. The research sample included 48 patients, treated with periodontal therapy and followed through a supportive periodontal program with recall sessions every 3 to 6 months for the next 5 to 16 years. The radiographic data showed that progression occurred in 28% and 38% of the supracrestal and intrabony defects, respectively, with the difference being not statistically significant [7].

Similarly, Rams et al. [8] examined 56 patients, previously treated with non-surgical and surgical periodontal therapy and followed on a 3-month recall program for the next 30 months. The prevalence of intrabony defects was 5%, while disease progression was 14.7% in teeth with intrabony defects and 1.8% in teeth with horizontal bone loss. It is noteworthy that the involved teeth in this research were posterior teeth, which have a higher rate of recurrent disease activity [8].

## 2. Surgical Treatment of Intrabony Defects

Treatment protocols for intrabony lesions were initially based on resective surgical procedures aiming at their elimination. With time more conservative techniques were introduced. It is well documented that non-surgical therapy is a precursor to surgical treatment. However, the non-surgical treatment of intrabony defects has rarely been used as monotherapy, and therefore, long-term data regarding its effectiveness are scarce. The limited available evidence demonstrated clinical attachment level (CAL) gains of 0.8 to 1.6 mm and minimal bone fill [9,10,11]. Open flap debridement (OFD) has shown a mean CAL gain of 1.6 mm and mean bone fill of 1 mm [12,13]. However, intrabony defects have been proven to exhibit high regenerative potential. One hundred and seventy-one of the 180 intrabony defects presented complete bone fill when treated with OFD and professional plaque removal was conducted biweekly for 2 years [14,15].

In order to increase the regenerative potential of intrabony lesions, regenerative techniques were used as adjunctive to surgical procedures. The development of periodontal regeneration in the last 30 years has followed two distinct, but totally different paths. For many years, the interest of researchers has been focused on regenerative materials and products, mainly exploring three different concepts: barrier membranes, bone grafts, biologic factors, and their combinations [16]. Bone grafts in the treatment of intrabony defects demonstrated an additional CAL gain of 1 mm in comparison to OFD; however, the results showed high heterogeneity [17,18]. Guided Tissue Regeneration (GTR) with the usage of resorbable or non-resorbable membranes seemed to enhance the results of the surgical treatment. Mean CAL gain of 3.4 mm has been reported with GTR procedures, compared to a mean 1.8 mm gain with OFD [19]. Similar favorable results were reported with the addition of Enamel Matrix Derivative (EMD) to surgical protocols. A systematic review mentioned an additional CAL gain of 1.1 to1.6 mm [20], while Koop et al. [21], in a meta-analysis calculated a mean CAL gain of 3.63 mm for the EMD treatment protocols, in comparison to a mean 1.91 mm gain without regenerative factors.

On a different path, the research community explored and focused on investigating novel surgical approaches. Clinical innovations in flap design and handling as well as minimally invasive approaches have radically changed regenerative surgery. Less invasive flap designs and clinical procedures with the use of papilla preservation flaps [22,23], was developed along with microsurgical instruments that were used to minimize periodontal tissue trauma and enhance clot stabilization [24].

The application of such minimally invasive techniques led to improved outcomes in comparison to conventional surgical techniques [25,26,27]. These techniques were initially presented by Harrel and Rees [28] in 1995 who introduced Minimally Invasive Surgery (MIS) and in 1999 presented a case series with intrabony defects treated with MIS in combination with Demineralized Freeze-Dried Bone Allograft (DFDBA) and vicryl mesh membrane [29]. The MIS technique included intrasulcular incisions around the involved teeth and a small incision at a distance of 2–3 mm from the top of the interdental papilla buccally, or palatally if it was an aesthetic area. The flap incision was performed with an Orban knife, the size of which was 1/3 or 1/4 of the regular size. Subsequently, the granulation tissue was completely removed from the pocket with special rotary instruments, the root surface was smoothened with burs, grafts and membrane were applied and the flap was sutured with vertical mattress sutures. The surgical procedure was performed with ×3.5 magnification.

Through the following years, many research groups presented similar techniques with minor differences in flap design. Cortellini and Tonetti presented in 2007 the Minimally Invasive Surgical Technique (MIST) [26]. The surgical procedure was according to a modified [22] or simplified [23] papilla preservation technique, presented by the same authors in 1995 and 1999, respectively. MIST included the use of mini curettes and thin ultrasonic tips for the removal of hard deposits and granulation tissue. In 2009, the authors suggested an even more minimal flap design the Modified-Minimally Invasive Surgical Technique (M-MIST) [27]. The difference between these techniques was that in the M-MIST only a minimal buccal flap was elevated, and granulation tissue removal was achieved without separation of the soft tissue from the bone. In both MIST and M-MIST techniques, EMD was additionally applied as a regenerative factor.

Concurrently, another research group presented the Single-Flap Approach (SFA) [30], while in 2012 they modified this flap design for application in wider intrabony defects (Double Flap Approach—DFA) [31]. The results of such minimally invasive techniques were so favorable that the additional need and benefit of grafting materials or biologics came into question. M-MIST was compared to M-MIST with the addition of a xenograft, with the results presenting no statistically significant differences [32]. Similarly, MIST was examined as a monotherapy compared to a combined approach with the addition of EMD, and again no significant differences between the two groups were found [33]. Trombelli et al. [34] also compared the SFA technique to SFA with the additional application of Hydroxyapatite (HA) and the use of a collagen membrane. The authors concluded that regenerative materials did not seem to enhance the clinical outcomes of the SFA technique. In a review published on minimally invasive surgical techniques [35] the advantages of such techniques were highlighted, stating that the interdental soft tissues act like a stable “roof”, preventing the loss of volume and contributing to blood fill and clot formation and stabilization, thus preventing the need for additional application of regenerative materials.

## 3. Minimally Invasive Non-Surgical Technique

To further reduce the need for surgical approaches in the treatment of intrabony defects, Ribeiro et al. [36] proposed the minimally invasive non-surgical technique (MINST)]. The non-surgical approach, the use of magnification for better visualization, and the application of less invasive instruments to avoid tissue trauma were hypothesized to lead to favorable outcomes for the treatment of intrabony defects and simultaneously avoid the unpleasant outcomes of surgical approaches, such as the psychologic burden for patients, surgical morbidity or esthetic complications. The study compared the non-surgical to a surgical minimally invasive technique in both clinical and patient-centered outcomes. Twenty-seven patients were treated in this randomized clinical trial and followed for 6 months. The MINST procedure consisted of scaling and root planing with mini curettes and ultrasonic instrumentation with thin and delicate tips and the use of an operative microscope. During scaling, instruments were carefully inserted through the periodontal pocket in order to preserve the stability of the soft tissues. No significant differences between surgical and non-surgical minimally invasive approaches were found. Pocket depth reduction was 3.51 mm for the MIST group and 3.13 mm for the non-surgical group and CAL gain was 2.85 mm and 2.56 mm, respectively. Patient pain and discomfort during the procedure, as well as esthetic satisfaction 6 months after treatment, did not present any significant differences. The only difference observed was the chair-time, which was significantly lower in the non-surgical group. At the 12-month follow-up examination, the results were still comparable between the two treatment procedures. Moreover, microbiological results in the defects demonstrated a significant reduction in the red complex species-P. gingivalis and T. forsythia—from baseline to 3 months, and maintenance throughout 12 months in both groups [37]. Thus, it was concluded that the non-surgical approach could yield comparable results to a surgical approach, with all the benefits that were previously mentioned coming into focus.

The first study that presented radiographic data of the MINST technique was published by Nibali et al. [38]. This was a retrospective study including 23 patients with 35 intrabony defects. MINST was applied in all defects with a magnification of ×3.4 and mini-Gracey curettes and piezo-electric devices with thin and delicate tips were used to minimize soft tissue trauma. Gingival curettage and “smoothening” of the root surface were avoided. Subgingival irrigation was also avoided to enhance the formation of a stable blood clot. Clinical and radiographic outcomes 12 months after treatment showed 3.12 mm pocket depth reduction, 2.78 mm CAL gain, and 2.93 mm intrabony depth reduction. The authors examined the factors affecting the outcomes and concluded that the deeper and narrower the lesion, the better the outcome. It was demonstrated that for defects wider than 45° the treatment outcome was poorer. Three years later, the same research group published long-term data with a 5-year follow-up of 14 patients, concerning 21 defects. The results showed stability of the radiographic data and additional minor clinical improvements. This study documented for the first time that the MINST approach is a long-term predictable therapeutic approach for the treatment of intrabony lesions.

Following these initially favorable results, MINST was combined with regenerative factors, and EMD was the material of choice. Non-surgical treatment of intrabony defects with the additional application of EMD was previously investigated without ever presenting any significant clinical improvements compared to non-surgical treatment alone [39,40,41]. In 2003, Gutierrez et al. [39], compared non-surgical periodontal therapy and non-surgical periodontal therapy with the adjunct application of EMD for the treatment of intrabony defects. Twenty-two untreated periodontal patients with bilateral defects were treated and followed for 3 months. No difference occurred between groups regarding pocket depth reduction (2.3 mm for the control group and 2 mm for the EMD group) and CAL gain (1.8 mm for the control group and 1.4 mm for the test group). Similarly, Mombelli et al. [40] reported for the non-surgical plus EMD group CAL gain of less than 0.5 mm, while Sculean et al. [41] mentioned a CAL gain of 1.5 ± 1.3 mm (ultrasonic instruments plus EMD group) and 2.0 ± 0.7 mm (hand periodontal instruments plus EMD). In this research, Sculean et al. [41] also presented histological data on the defects. Six months after treatment, the involved teeth were extracted, and histological analysis was performed. The authors mentioned that in only 4 of the 10 teeth in which EMD was applied, new cementum was identified (0.2–0.6 mm) and only 2 out of 10 exhibited new bone formation (0.2–0.3 mm). An important parameter is that the teeth in this research were hopeless teeth, scheduled for extraction, with 11–13 mm clinical attachment loss limiting the effectiveness of the non-surgical therapy. It is well established that when the pocket depth increases, the effectiveness of hard deposit removal is reduced [42]. Periodontal pockets deeper than 6 mm, have been found to exhibit hard deposits in 19–66% of the root surface [43].

In contrast to the previous findings, encouraging clinical and histologic outcomes of the combination of scaling and root planing with the additional application of EMD were presented in 2009. Mellonig et al. [44] published a histologic case series of four subjects, where four hopeless teeth with deep intrabony defects were treated. Non-surgical periodontal therapy with gracey and ultrasonic instruments was performed and EMD was applied in all defects. All teeth were extracted 6 months after treatment and histologic evaluation was performed. The results demonstrated that new cementum (0.42–2.88 mm), new bone (1.07–3.21 mm), and new periodontal ligament were found in three out of four defects. This was the first publication that confirmed that enamel matrix derivative (EMD) may lead to significant regenerative outcomes when used as an adjunct to non-surgical periodontal therapy. In the discussion of the article, the authors mentioned that these positive results could have been attributed to the intense scaling and root planing which was performed without any time limitations, and maybe lead to the complete removal of the periodontal fibers in the bottom of the defects and subsequently an improved proliferation of the periodontal ligament cells. The EMD application may have played an important role in the significant periodontal regeneration presented, preventing the intrusion of epithelial cells and giving time to the other tissues to develop in the defect area. Moreover, the application of EMD has been well-documented to promote cementum and bone regeneration [45]. These favorable regenerative outcomes of EMD application led the researchers to re-examine the addition of EMD in a more conservative non-surgical procedure (MINST).

Aimetti et al. [46] published a case series of 11 patients with a 2-year follow-up of MINST and adjunctive application of EMD. During the procedure, a gingival retractor was used with magnification of ×12.5 along with a micro-surgical dental mirror to accomplish better visualization. Mini curettes and an ultrasonic device with thin and delicate tips were used to minimize tissue trauma. When no more calculus could be detected, EDTA gel was applied to the bottom of the defect for 2 min. The area was rinsed with saline and EMD application was performed. Clinical results showed a 3.6 mm pocket depth reduction and 3.2 mm CAL gain. The reduction in the intrabony depth, measured radiographically, was 2.7 mm. One year later, the same research group presented a randomized clinical trial comparing the efficacy of MINST and minimally invasive surgical technique with the additional application of EMD in both groups [47]. The surgical technique performed was either SFA [30] or M-MIST [27]. The 24-month outcomes of the 30 participants of the study did not differ significantly between the two treatment protocols. The MINST group showed a 3.6 mm pocket depth reduction with 3.2 mm CAL gain, while in the surgical approach, the results were 3.7 mm and 3.6 mm, respectively. The difference was statistically significant only in radiographic measurements. Intrabony depth reduction was 2.6 mm in the flapless approach and 3.8 mm in the surgical approach. Patient-centered outcomes showed that pain and discomfort during therapy, as well as one week post-operatively was not statistically significant between groups, whereas chair time was significantly reduced in the flapless approach.

After this initial documentation on the efficacy of MINST therapy for the treatment of intrabony defects, the question that arose was if using additional regenerative factors would improve the outcomes. Thus, the next publication concerning MINST investigated the difference between MINST and MINST with the additional application of EMD [48]. The MINST procedure was performed using specialized instruments, mini curettes, and piezo-electric devices with thin and delicate tips. Root surface visualization was enhanced with a magnification of 3× and fiber optic lighting. In the MINST + EMD group, EDTA gel was applied to the bottom of the defect for 2 min after the completion of the MINST procedure, rinsed thoroughly with saline and the root surface was dried. EMD was applied on the bottom of the defect with gentle compression to maintain soft tissue stability (Figure 1).

The results of this study demonstrated no statistically significant differences in both mean clinical and radiographical parameters 12 months after treatment. Pocket depth reduction was 4.0 mm and CAL gain was 3.5 mm in the MINST group. In the MINST + EMD group the treatment outcomes observed were 4.2 mm and 3.4 mm, respectively. Radiographic measurements indicated a 1.9 mm intrabony depth reduction for the MINST and 1.8 mm for the MINST + EMD group.

Regarding clinical outcomes, it is noteworthy that the application of EMD led to significantly fewer residual pockets with PD ≥ 5 mm in the 12-month follow-up (33.3% in the MINST vs. 16.6% in the MINST + EMD group). Moreover, pocket closure (PD ≤ 4 mm with no BoP) was also achieved in more defects of the MINST + EMD group compared to the MINST group (77.8% vs. 55.6% sites), particularly for sites with baseline PD ≤ 8 mm (92.3% vs. 69.2%). In this research, the authors also evaluated a new treatment outcome, introduced by Trombelli et al. in 2020 [49], entitled “Composite Outcome Measurement (COM)”. This parameter consisted of a combination of clinical outcomes validating a successful therapeutic result (PD ≤ 4 mm and CAL gain ≥ 3 mm). The 1-year results demonstrated an increased number of successful COM (61.1% vs. 44.4% of sites). The enhanced pocket reduction and pocket closure observed with the application of EMD have also been reported in other studies where EMD was applied during initial non-surgical therapy [50], or during re-instrumentation of residual pockets persisting after steps 1 and 2 of periodontal therapy [51]. Graziani et al. [52], also claimed that a lower number of cases with residual PPD ≥ 6 mm was observed at 3 months, when EMD was applied in conjunction with non-surgical therapy.

Another advantage of the MINST is the reduced time duration of the treatment. According to the published studies, the MINST application lasted less than a surgical therapy session but more than a conservative scaling and root planing session. Ribeiro et al. [36] mentioned that the time of MINST application was 29.15 min, while Aimetti et al. [47] reported a duration of 23.5 min for the MINST group. The results were similar in the Anoixiadou et al. [48] trial. The MINST group was treated within 22.5 min, and the time for the MINST + EMD group was 25.3 min.

## 4. Conclusions

In a 2015 consensus report, surgical approaches were mentioned to be the treatment of choice for intrabony defects [53]. Moreover, MINST procedures had the lowest probability to be the best treatment as reported in a recent network meta-analysis [54]. However, recent comparisons of minimally invasive non-surgical with minimally invasive surgical techniques, have exhibited similar favorable outcomes for both approaches, with or without the additional application of biologics [36,37,47]. The authors of the aforementioned meta-analysis [54], comment in their discussion that clinical differences in terms of CAL gain between MINST and MIS procedures are small and considering the significantly higher treatment chair time for MIS procedures, further investigations are needed to explore cost-benefit ratio MIS and MINST.

The application of MINST minimizes soft and hard tissue trauma achieving better treatment outcomes in comparison with traditional scaling and root planing. Gingival architecture is maintained, and stability of the area is preserved favoring better clot formation and stabilization. Thus, the healing potential of the defect is enhanced. The use of biologics in non-surgical periodontal therapy is being investigated since the early 2000s in multiple research centers. The addition of EMD, despite the well-documented favorable results in composite outcomes, to the MINST procedure did not seem to significantly contribute to the mean clinical and radiographic improvements (Table 1). Considering the limited number of published studies regarding the efficacy of MINST, more investigations are necessary, including more patients and centers, to confirm its effectiveness with or without the application of EMD.

The main limitation of a non-surgical versus a surgical approach is the reduced visualization with the closed instrumentation which may lead to the inability to achieve complete removal of hard deposits. Even though scientific documentation is still missing, current advances in visualization through magnification and fiber-optic lighting usage may overcome this problem. Careful evaluation of the location and extent of the intrabony defect is necessary before applying MINST. Anterior defects, interproximal defects without buccal and lingual extension, 3/2-wall defects, and defects with probing depth < 8 mm that facilitate access and possibly enhance the effectiveness of close instrumentation are preferred.

Moreover, a limitation of all the current studies with MINST is that the exact morphology of the bony defects could not be determined due to the non-surgical nature of the treatment, thus preventing further analysis of the influence of defect configuration in the outcome. It is well-documented that defect morphology, angle, and number of bony walls of the intrabony defect, are important factors in surgical regenerative outcomes [55]. Unfortunately, data regarding the influence of defect configuration on the MINST approach has not yet been presented. More studies including preoperative 3D radiographic evaluation are needed in order to specify the exact beneficial effect of the MINST and possible limitations in relation to the intrabony depth, width, and number of bony walls of the defect which will determine if the MINST or a surgical approach is the preferred treatment of choice. However, there is no doubt that contemporary treatment of intrabony defects is moving towards less invasive approaches and improved well-being of the patients.

## Figures and Tables

**Figure 1 dentistry-11-00025-f001:**
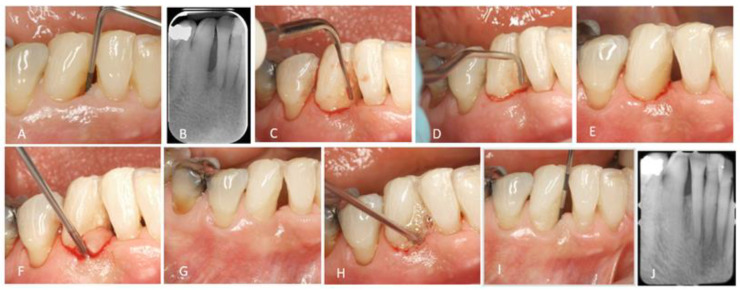
MINST + EMD application: (**A**,**B**) Initial clinical and radiographic images; (**C**) Use of ultrasonic delicate tip; (**D**) Use of mini curette; (**E**) After MINST (notice the minimal tissue trauma); (**F**) EDTA application; (**G**) Dried tooth surface; (**H**) EMD application; (**I**,**J**) Clinical and radiographic images at 12 months.

**Table 1 dentistry-11-00025-t001:** Results of MINST and MINST + EMD.

Authors	Follow-Up	Treatment Procedure	PD Reduction	CAL Gain	Intrabony Depth Reduction
Ribeiro et al., 2013 [38]	12 months	MINST	3.19 mm *	2.58 mm *	-
Nibali et al., 2015 [39]	12 months	MINST	3.12 mm *	2.78 mm *	2.93 mm *
Anoixiadou et al., 2021 [48]	12 months	MINST	4.0 mm *	3.5 mm *	1.9 mm *
Aimetti et al., 2017 [47]	12 months	MINST + EMD	3.4 mm *	3.1 mm *	2.1 mm *
Anoixiadou et al., 2021 [48]	12 months	MINST + EMD	4.2 mm *	3.4 mm *	1.8 mm *

* statistical significance from baseline < 0.05.

## Data Availability

No new data were created or analyzed in this study. Data sharing is not applicable to this article.

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
