# Peer review of "Minimally Invasive Non-Surgical Technique in the Treatment of Intrabony Defects—A Narrative Review"

_dentistry, 2023, doi:10.3390/dj11010025_

Round 1

Reviewer 1 Report

Dear Authors,

It was good to see a narrative review to put all the existing evidence on this topic together for the readers. But the conclusion needs to be well-supported by the evidence. Since the existing evidence still is lacking in terms of the comparison of MIST versus MINST, the authors should be cautious about how the message is delivered.

General comments:

1.     Page 7, the “Composite Outcome Measurement (COM) proposed by Trombelli et al (2020) is a combination of PD≤ 4mm (pocket closure) and CAL gain ≥ 3mm.

“gain” was missing in the context.

2.     The conclusion drawn in this manuscript “comparisons of minimally invasive non-surgical with minimally invasive surgical techniques, have exhibited similar favorable outcomes for both approaches” is not well-supported by the existing evidence.  Only Ribeiro et al. two articles were available, not including Aimetti et al. . It was discussed in the conclusion that the result of the comparison may vary among different preoperative pocket depths.  

Table: Please put the statistical significance in the table.

References:

1.     #28 reference needs to be re-edited to be consistent with the citation in the text (Harrel & Rees, 1995)

2. The #48 reference should be removed from the second sentence of the conclusion. 

Author Response

  1. Page 7, the “Composite Outcome Measurement (COM) proposed by Trombelli et al (2020) is a combination of PD≤ 4mm (pocket closure) and CAL gain ≥ 3mm. “gain” was missing in the context.

Thank you for your comment. The word “gain” has been added. (Page 7, line 309)

  1. The conclusion drawn in this manuscript “comparisons of minimally invasive non-surgical with minimally invasive surgical techniques, have exhibited similar favorable outcomes for both approaches” is not well-supported by the existing evidence. Only Ribeiro et al. two articles were available, not including Aimetti et al. . It was discussed in the conclusion that the result of the comparison may vary among different preoperative pocket depths. 

Aimetti et al. compared the outcomes of flapless approach to minimally invasive surgical technique with the additional application of EMD in both groups. Considering that the addition of the regenerative material is the same in both groups, the possible difference between groups will be attributed to the different technique. We added “with or without the additional application of biologics [38- 39],[48].” (Page 8, line 328)

We agree with the reviewer, we added the phrase “However, the evidence on this comparison is still scarce and more researches are needed to confirm this hypothesis.” (Page 8, line 329)

  1. Table: Please put the statistical significance in the table.

Thank you for your comment, the table has been corrected.

  1. References:
  2. #28 reference needs to be re-edited to be consistent with the citation in the text (Harrel & Rees, 1995)
  3. The #48 reference should be removed from the second sentence of the conclusion.

Thank you for your comment. Reference #28 has been corrected (Page 4, line 135).

Reference #48 has been renumbered.

Reviewer 2 Report

Dear authors,

below my comments on your paper. A narrative review on Minimally invasive non-surgical technique in the treatment of IB defects.

From the title you stated that the focus of this NR was the MINST. However, you did a resume of the evolution of regenerative approaches to IB defect and only the final part of the manuscript was specific for MINST.

In literature there is strong evidence showing that non-surgical treatment is characterized by residual calculus. You discussed this very important issue but I suggest you to improve this section.

In my opinion the evidence supporting MINST is “young” and not conclusion could be drawn.  

In fact, in a recent SR with network MA (doi: 10.1007/s00784-020-03229-0) was reported that both MIS and MINST are promising for the treatment of residual pocket with IB defect. However, MINST showed the lowest probability to be the best treatment for CAL Gain.

The use of biologics without a perfect debridment of the root is something that should be carefully considered, also from an ethical point of view.

There are errors/mismatch in references number reported in the manuscript and in the reference list respectively.

Regards

Author Response

  1. From the title you stated that the focus of this NR was the MINST. However, you did a resume of the evolution of regenerative approaches to IB defect and only the final part of the manuscript was specific for MINST.

Thank you for your comment. We believe that a review of evolution of regenerative therapy is necessary for the readers to understand this issue. However, we agree with the reviewer that the review is too extensive. We submitted a shorter version of the article, but the editorial board informed us that we have to have a certain number of words for the article to be accepted. If this word limit is not a prerequisite for article publication, we can submit the shorter version again.

  1. In literature there is strong evidence showing that non-surgical treatment is characterized by residual calculus. You discussed this very important issue but I suggest you to improve this section.

Thank you for your comment. We added “Careful evaluation of the location and extent of the intrabony defect is necessary before applying MINST. Anterior defects, interproximal defects without buccal and lingual extension, 3- 2- wall defects, and defects with probing depth < 8mm that possibly enhance the effectiveness of calculus removal during close instrumentation are preferred.”

Moreover, the issue of possible residual calculus present in defects treated with MINST, did not seem to be detrimental to the clinical outcomes, as evidence by the results of all published studies on this treatment modality.

  1. In my opinion the evidence supporting MINST is “young” and not conclusion could be drawn.  In fact, in a recent SR with network MA (doi: 1007/s00784-020-03229-0) was reported that both MIS and MINST are promising for the treatment of residual pocket with IB defect. However, MINST showed the lowest probability to be the best treatment for CAL Gain.

Considering the limited number of published studies regarding the efficacy of MINST, more investigations are necessary including more patients and centers to confirm its effectiveness with or without the application of EMD. (Page 8, line 337)

  1. The use of biologics without a perfect debridment of the root is something that should be carefully considered, also from an ethical point of view.

We would like to point out that this article is a review of studies that have been published in highly esteem periodontal journals. It becomes obvious that all of these studies have received approval from the relevant ethics committees, which is a prerequisite for their publication. Therefore, we believe that the ethical considerations that you have commented upon are not relevant to the scope of this review paper.

The use of the biologics in non-surgical periodontal therapy is being investigated since the 2000s in multiple research centers of high renown (Page 9, line 354).

  1. There are errors/mismatch in references number reported in the manuscript and in the reference list respectively.

Thank you for your comment. References have been corrected.

Round 2

Reviewer 1 Report

Dear Authors,

All the concerns are addressed. Thank you!

Author Response

Thank you vey much.

Reviewer 2 Report

Dear Authors,

thank you for your efforts in improving your manuscript.

Honestly, I disagree that residual calculus cannot be detrimental in the long term. 

Please, as I suggest before, add the statement that MINST procedures had the lowest probability to be the best treatment as reported in a recent network meta-analysis.

This is just my opinion, ethic approval from a committee doesn't exempt us from ethics consideration.

Regards

Author Response

Dear Sir/Madam,

Thank you once again for your comments.

The article is added in the reference list (54). The statement that you suggested and minor analysis of that article were added in page 8, (l. 326-327, l.330-334) and noted with yellow line.

Thank you in advance,

Stella Anoixiadou

Round 3

Reviewer 2 Report

Dear authors, 

thank you for your efforts.

Regards